# ONLINE BIAS CORRECTION FOR TASK-FREE CONTINUAL LEARNING

**Aristotelis Chrysakis & Marie-Francine Moens**[*]
Department of Computer Science
KU Leuven
Leuven, Belgium

## ABSTRACT

Task-free continual learning is the machine-learning setting where a model is trained online with data generated by a nonstationary stream. Conventional wisdom suggests that, in this setting, models are trained using an approach called experience replay, where the risk is computed both with respect to current stream observations and to a small subset of past observations. In this work, we explain both theoretically and empirically how experience replay biases the outputs of the model towards recent stream observations. Moreover, we propose a simple approach to mitigate this bias online, by changing how the output layer of the model is optimized. We show that our approach improves significantly the learning performance of experience-replay approaches over different datasets. Our findings suggest that, when performing experience replay, the output layer of the model should be optimized separately from the preceding layers.

## 1  INTRODUCTION

In broad terms, *continual learning* is the process of incrementally aggregating knowledge from data that are generated by a nonstationary distribution (Lee et al., 2019; Riemer et al., 2019). The main motivation for studying continual learning is to give artificial learners the ability to learn as biological learners do—perpetually updating and refining their body of knowledge under changing external conditions (Silver et al., 2013). The inability of artificial learners to learn continually stems from the fact that they overwrite previously learned knowledge whenever they encounter new information. This phenomenon is called *catastrophic forgetting* (McCloskey & Cohen, 1989; French, 1999).

In this paper, we focus specifically on *task-free* continual learning (Aljundi et al., 2019b). In this setting, the data are presented to the learner in small minibatches, and this setting is agnostic to the way the data distribution changes over time. In other words, we assume no knowledge about whether the distribution is piecewise-stationary (that is, when there are distinct tasks being learned), or whether the distribution changes continuously over time (Aljundi et al., 2019b). Most task-free continual learning approaches make use of a memory which can store a small percentage (typically 10% or less) of all observed data instances. The data instances stored in memory are subsequently replayed in order to mitigate catastrophic forgetting. This simple paradigm, called *replay-based* continual learning, is surprisingly effective in task-free settings. Furthermore, it is also supported by findings from the field of neuroscience, in relation to how biological learning takes place (Marr, 1971; Ji & Wilson, 2007; Liu et al., 2019).

A number of continual learning methods tend to make predictions that are biased towards recently observed data (Buzzega et al., 2021; Mai et al., 2021). Several strategies have been proposed to deal with this prediction bias (also called recency bias). Unfortunately, most of them are not applicable to task-free continual learning, since they have been designed for continual learning settings that consist of a task sequence, and they require knowledge of which classes the current task comprises (Wu et al., 2019; Belouadah & Popescu, 2019; Buzzega et al., 2021). One approach which is applicable in task-free continual learning is proposed in Mai et al. (2021), but it can only be performed after the end of training, hence the learner's predictions *during* training would remain biased.

---

[*]Please address your correspondence to aristotelis.chrysakis@kuleuven.be.

In this paper, we propose a simple approach that performs online bias correction for task-free continual learning. Our contributions are as follows: a) We formally illustrate that the conventional paradigm of model training in task-free continual learning overweights the importance of current stream observations (Section 3.2), and we speculate that this overweighting is a cause of prediction bias of continual learners; b) We propose a novel metric to quantify prediction bias (Section 3.3), and we show that this bias can be effectively mitigated by appropriately modifying the parameters of only the final layer of the model, after the end of training (Section 3.4); c) We propose a novel approach called Online Bias Correction (OBC; Section 3.5), which maintains an unbiased model online, throughout the entire duration of learning (see Figure 1 for an illustration); d) We evaluate the performance of OBC extensively, and we show that it significantly improves a number of task-free continual learning methods, over multiple datasets (Section 4).

## 2 BACKGROUND

### 2.1 TASK-FREE CONTINUAL LEARNING

We define *task-free* continual learning as the online optimization of a model via small minibatches that are sampled from a nonstationary stream. In task-free continual learning, no strong assumptions are made about the nature of the distributional nonstationarity of the stream (Aljundi et al., 2019b). Other continual learning settings, such as *task-incremental* and *class-incremental* continual learning assume a data distribution that is piecewise stationary, hence one that only changes at discrete points in time (Van de Ven & Tolias, 2019). The objective of continual learning is to learn from *all* observed data despite the nonstationary nature of the distribution (Jin et al., 2021), and, in general, previous work assumes no distributional mismatch between training and evaluation data.

Previous work in task-free continual learning mostly focuses on *replay-based* methods (Aljundi et al., 2019c; Jin et al., 2021). The prevalent replay paradigm is called *experience replay* (ER) (Isele & Cosgun, 2018; Chaudhry et al., 2019). According to the ER paradigm, each minibatch of observations received by the learner is combined with another minibatch of equal size sampled from the memory. The model is then trained for one step with the combined stream-and-memory minibatch. Moreover, the memory is typically maintained by an online memory-population algorithm called reservoir sampling (Vitter, 1985).

There are multiple variants of the ER method. For instance, one approach called Maximally-Interfered Retrieval (MIR) replays instances that are going to be interfered the most by the current minibatch of new observations. Another approach called Class-Balancing Reservoir Sampling (CBRS) (Chrysakis & Moens, 2020) modifies the memory population algorithm to ensure that the memory remains balanced. There also other approaches that deviate from the ER paradigm, such as Greedy Sampler and Dumb Learner (GDUMB) (Prabhu et al., 2020), which only trains the model using data stored in memory, or Asymmetric Corss-Entropy (ACE) (Caccia et al., 2022), which uses a modified version of the cross-entropy loss to prevent the drift of latent representations.

### 2.2 COMPUTATIONAL COST

An important issue in task-free continual learning is computational cost. Since practical applications will likely involve large amounts of data, the design of task-free continual learners should ensure they are tractable. In practical terms, let us assume that a model has to learn from a stream of $n$ instances. Moreover, we assume that applications with larger streams will likely involve memory storages of larger size $m$. In real-world applications, the difference between an $O(n)$ learning algorithm, and an $O(mn)$ algorithm could be enormous. Hence, in this work we will only focus on learning algorithms whose computational cost per incoming batch is independent of the memory size $m$, so that the computational complexity of learning from the entire stream is $O(n)$.

### 2.3 BIAS CORRECTION IN TASK-FREE CONTINUAL LEARNING

To the best of our knowledge, there is only one approach explicitly designed to correct for prediction biases in task-free continual learning. Mai et al. (2021) propose learning a model using conventional experience replay, and after the entire stream has been observed, they replace the final linear layer of the model with a nearest-class-mean (NCM) classifier computed using all data stored in mem-

ory. Moreover, they demonstrate that this approach is effective in increasing the final accuracy of the model. However, in many real-world applications, there is a need for a model that learns and performs inference at the same time, and such a model should ideally be unbiased all the time. To achieve this goal, the NCM approach would have to be applied after every update of the model, and since it needs to make a full pass over the memory, it would be computationally very expensive.

# 3 METHODOLOGY

## 3.1 NOMENCLATURE

Previous work in continual learning typically views the entire neural network as one learning component. In contrast, we adopt a more modular view of a neural network that consists of two learning components. Specifically, we will call the output layer of a neural network the *classifier*, and we will denote it as the parameterized function $c(\boldsymbol{z}; \boldsymbol{\theta}_c)$. Moreover, we will refer to the set of layers preceding the classifier as the *feature extractor*, and we will similarly denote it as $g(\boldsymbol{x}; \boldsymbol{\theta}_g)$. Using this notation, we can represent the full neural network $h$ as the composition of $g$ and $c$, that is

$$h(\boldsymbol{x}; \boldsymbol{\theta}_h) \triangleq c\big(g(\boldsymbol{x}; \boldsymbol{\theta}_g); \boldsymbol{\theta}_c\big), \quad \text{where } \boldsymbol{\theta}_h \triangleq \{\boldsymbol{\theta}_g, \boldsymbol{\theta}_c\}. \tag{1}$$

An important distinction between the classifier and the feature extractor, is that classifier has a low learning capacity because it consists of only one linear layer, while the feature extractor has a high learning capacity because it is composed of multiple nonlinear layers. Hence, given enough data, the feature extractor can learn more complex representations, but, in low-data scenarios, it is also more prone to overfitting, in comparison to the classifier.

## 3.2 DATA-SAMPLING BIAS

At a high level, the optimization process during task-free continual learning is very simple. At each step $t$, the learner receives a small minibatch of $b$ observations $\mathbf{S}_t = \{(\boldsymbol{x}_i, y_i)\}_{i=1}^b$ from the stream. The learner then samples another minibatch $\mathbf{R}$ of equal size from its memory $\mathbf{M} = \{(\boldsymbol{x}_i, y_i)\}_{i=1}^m$, and performs an update step over the model parameters using the gradient calculated with respect to $\mathbf{S}_t \cup \mathbf{R}$. Finally, the learner updates its memory with respect to $\mathbf{S}_t$, using reservoir sampling (Vitter, 1985), or another memory population algorithm. This training paradigm is called *experience replay* (Isele & Cosgun, 2018; Chaudhry et al., 2019). We will now explain why this paradigm has a data-sampling bias that favors new observations over the instances stored in memory.

An intuitive way to understand this data-sampling bias is to consider a learner with an infinite memory. Let us assume that the learner has encountered $a$ data instances in the past—all of which have been stored in the memory for replay—and that the learner now receives a minibatch $\mathbf{S}_t$ of $b$ new observations.[1] Given an Occam's-razor assumption that all observations, current and past, are equally important, we will say that the data-sampling is *unbiased* if all $a + b$ observed data instances are equally likely to contribute to the upcoming optimization step of the model. In other words, the probability of using any instance in the optimization step should be the same, regardless of whether that instance is in the current stream minibatch $\mathbf{S}_t$, or stored in memory. Accordingly, the model should be updated with a minibatch $\mathbf{B}$ sampled uniformly-at-random from the set of all $a + b$ observed instances (that is, the concatenation of the memory and the stream minibatch $\mathbf{M} \cup \mathbf{S}_t$).

Now let us contrast this unbiased data sampling with experience replay. Under experience replay, the $b$ new observations are included in the minibatch $\mathbf{B}$ with a probability of 1, and, by definition, an equal number of instances are sampled uniformly-at-random from the memory (which contains $a$ instances). Hence, each memory instance has a probability of $b/a \ll 1$ of being sampled for the model update. Therefore, unlike in the unbiased case described above, new observations are guaranteed to participate in the model update, but an arbitrary memory instance is very unlikely to. In essence, this is a data-sampling bias that favors current observations over past ones, and, in turn, leads to the predictions of the model being biased towards recent observations.[2]

---

[1]Since the new observations arrive in small minibatches, we generally assume that $a \gg b$. Put another way, the instances in memory vastly outnumber the instances in the newly observed minibatch.

[2]To further elucidate the concept of data-sampling bias, we include a numerical example in the appendix.

Table 1: We compare the final accuracy (Acc.) and the bias of ER, MRO, and ER with post-training bias correction (ER+BC), over four datasets. All entries are $95\%$-confidence intervals over 15 runs.

|  | FashionMNIST | | CIFAR-10 | | CIFAR-100 | | tinyImageNet | |
|---|---|---|---|---|---|---|---|---|
|  | Acc. | Pred. Bias | Acc. | Pred. Bias | Acc. | Pred. Bias | Acc. | Pred. Bias |
| ER | $83.4 \pm 0.5$ | $0.9 \pm 0.2$ | $45.5 \pm 1.7$ | $17.0 \pm 2.4$ | $19.3 \pm 0.7$ | $33.8 \pm 2.0$ | $13.5 \pm 0.5$ | $25.3 \pm 1.7$ |
| MRO | $83.7 \pm 0.5$ | $0.4 \pm 0.3$ | $38.9 \pm 1.6$ | $8.7 \pm 2.7$ | $14.7 \pm 0.6$ | $17.0 \pm 2.2$ | $10.7 \pm 0.4$ | $12.6 \pm 1.3$ |
| ER+BC | $84.8 \pm 0.2$ | $0.4 \pm 0.1$ | $54.4 \pm 0.8$ | $6.6 \pm 0.5$ | $27.4 \pm 0.4$ | $6.3 \pm 0.4$ | $20.4 \pm 0.3$ | $5.1 \pm 0.3$ |

A hypothesis arising from this discussion is that prediction biases could be negated if the learner had a memory of infinite size, and if it constructed the minibatch $\mathbf{B}$, used to update the model, by sampling instances uniformly-at-random from all observed data, current or past. In this case, we would expect that, on average, a ratio of $b/(a + b)$ of the instances in the minibatch $\mathbf{B}$ will be from $\mathbf{S}_t$, and $a/(a + b)$ of the instances will be from $\mathbf{M}$. Unfortunately, in practical applications, it is not possible to have a memory of infinite size. As the learner observes more and more data, it is $a \gg b$, which means that $a/(a + b) \simeq 1$ and $b/(a + b) \simeq 0$. Therefore, on average, the minibatches $\mathbf{B}$ used to train the model will contain almost exclusively data from the memory, and almost no data from the current stream batch $\mathbf{S}_t$. Since in practical applications the size of the memory is much smaller than the size of the stream, this weighting scheme will arguably lead to overfitting the memory data.

### 3.3 QUANTIFYING PREDICTION BIAS

At this point, we will propose a metric to quantify prediction bias with respect to a set of unseen data. Let $\mathbf{T} = \{(\boldsymbol{x}_i, y_i)\}_i$ be this set, and by $\boldsymbol{y}_i$, we will denote the one-hot representation of the label $y_i$. We define the expected vector of prior class probabilities as $\boldsymbol{p} = \mathbb{E}[\boldsymbol{y}_i]$. Moreover, we define the expected vector of model predictions as $\boldsymbol{q} = \mathbb{E}[h(\boldsymbol{x}_i; \boldsymbol{\theta}_h)]$. Note that both expectations are taken with respect to the data distribution of $\mathbf{T}$, and are computed by averaging over the test-set instances.

We quantify the prediction bias of the model by measuring the discrepancy between the expected ground truth $\boldsymbol{p}$ and the expected prediction $\boldsymbol{q}$. Since both $\boldsymbol{p}$ and $\boldsymbol{q}$ are vectors of probabilities, we propose the use of the Jensen-Shannon divergence (Lin, 1991), which is a symmetric divergence measure between two probability distributions. It is defined as

$$\mathrm{JS}(\boldsymbol{p} \, || \, \boldsymbol{q}) = \frac{1}{2} \left[ \sum_{j=1}^{c} p_j \log \frac{\mu_j}{p_j} + \sum_{j=1}^{c} q_j \log \frac{\mu_j}{q_j} \right] \qquad (2)$$

where $\boldsymbol{\mu} = 1/2(\boldsymbol{p} + \boldsymbol{q})$ is the average of $\boldsymbol{p}$ and $\boldsymbol{q}$, and $j$ is the index over the classes $1, \ldots, c$, that are present in the test set. At a high level, when the model predictions are not biased, $\boldsymbol{p}$ and $\boldsymbol{q}$ will be very similar, and the divergence will be close to zero. Conversely, the more dissimilar $\boldsymbol{p}$ and $\boldsymbol{q}$ are, the more the divergence will increase.

### 3.4 POST-TRAINING BIAS CORRECTION

At this point, we will show that we can significantly reduce the prediction bias of a task-free continual learner if we appropriately change the parameter vector $\boldsymbol{\theta}_c$ of the classifier after the entire stream has been observed. We will compare three approaches over four datasets (we follow the experimental settings described in Section 4.1). We report both the accuracy and the prediction bias (Eq. 2) after the end of the stream. We compare regular experience replay (ER), which weights equally its present and past, and memory-replay only (MRO), which uses only data stored in memory to train the model. Moreover, we also evaluate a version of ER with post-training bias correction (ER-BC). To correct the bias of the model, we use the data stored in memory to train only the classifier of the model, until convergence. The feature extractor of the model remains unchanged.

The results of this comparison are presented in Table 1. We observe that MRO is less biased than ER in all four datasets. On the other hand, despite being more biased, ER outperforms MRO in CIFAR-10, CIFAR-100, and tinyImageNet. As we hypohtesized at the end of Section 3.2, training a model using only the data stored in memory will reduce the prediction bias of that model, but it could also

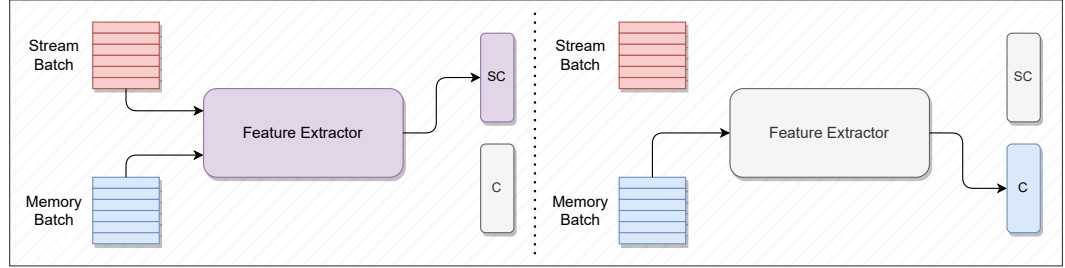

Figure 1: An illustration of Online Bias Correction (OBC). At first (left), the feature is extractor is trained using both a stream and a replay minibatch (as in conventional experience replay), with the help of a surrogate classifier (SC). Afterwards (right), the classifier (C) is trained using only a minibatch sampled from the memory, while the feature extractor remains unchanged. This two-step process is repeated for every incoming stream minibatch. The color of a model component illustrates the instances with respect to which its gradients are calculated (i.e., both stream and memory or only memory instances). Lack of color for a component means that it is not currently trainable.

lead to overfitting. On the contrary, ER+BC is less biased than ER, and achieves significantly higher accuracy values on all four datasets.

These results are evidence that prediction bias can be mitigated just by changing the way the classifier is trained. Moreover, because the classifier consists of just one layer, it has lower learning capacity, and, therefore, overfits much less than if we were to train the entire model using only the memory data (as is the case with MRO). In the following section, we exploit the findings of this section to propose an algorithm that maintains an unbiased model throughout the entire stream.

### 3.5 ONLINE BIAS CORRECTION

Our approach is called *Online Bias Correction* (OBC) and acts as a wrapper around other task-free continual learning methods that perform experience replay. At first, a generic task-free continual learning method performs its training step. Such a training step, typically includes receiving a minibatch of observations $\mathbf{S}_t$ from the stream (Line 1), sampling another minibatch $\mathbf{R}$ from memory (Line 2), and performing a training step using the combination of the two minibatches (Line 3). Note that only the parameter vector of the feature extractor (FE), and that of the surrogate classifier (SC) are updated during this training step. Next, the learner decides which of the new observations in $\mathbf{S}_t$ to store in memory, and which ones to discard (Line 4).

The following four steps (i.e., Lines 5-8) are the ones that OBC introduces. First, the feature extractor is frozen (Line 5)—that is, in the update step that follows, there will be no gradients with respect to its parameter vector calculated in the backward pass, and thus its parameter vector will not be updated. Next, a new minibatch $\mathbf{R}^*$ is sampled from the memory (Line 6). This memory minibatch is feedforwarded through the feature extractor and then the classifier (C), and only the parameter vector of the classifier, is updated with the resulting gradients (Line 7). Since the classifier is trained using only memory data, in order to reduce overfitting, we make use of label smoothing (Szegedy et al., 2016), which is a technique that adds noise to the ground-truth labels, in order to discourage overconfident predictions. Finally, the feature extractor is unfrozen, meaning that its parameter vector is trainable again (Line 8).

---

**Algorithm 1** Online Bias Correction

    Stream minibatches $\mathbf{S}_t, t = 1, 2, \ldots,$
    Memory $\mathbf{M}$, Feature Extractor (FE),
    Classifier (C), Surrogate Classifier (SC)

1: **for** each stream minibatch $\mathbf{S}_t$ **do**
2:     Sample a memory minibatch $\mathbf{R} \sim \mathbf{M}$
3:     Train the FE and the SC using $\mathbf{R} \cup \mathbf{S}_t$
4:     Perform memory population using $\mathbf{S}_t$
5:     Freeze the FE
6:     Sample a memory minibatch $\mathbf{R}^* \sim \mathbf{M}$
7:     Train the C using $\mathbf{R}^*$
8:     Unfreeze the FE
9: **end for**

---

The design of OBC was motivated by the preliminary experiments of Section 3.4. In particular, we exploit the fact that, as we saw earlier, it is possible to mitigate prediction bias by just optimizing the

weights of the classifier with respect to the data in memory. Moreover, the classifier, in the context discussed here, consists of a single layer, hence it is more resistant to overfitting compared to the feature extractor, which is typically a deep neural network of high learning capacity. Therefore, by training the classifier only using memory data, we mitigate the data-sampling bias, and hence, its prediction bias. Regarding the feature extractor, as we saw in Section 3.4, it is beneficial for it to be trained via experience replay, possibly because this approach leads to less overfitting compared to training the feature extractor only with memory data. Therefore, we introduced a surrogate classifier in order to ensure that the feature extractor is trained in exactly the same way as in experience replay (that is, in combination with a biased classifier). We have also validated this design choice experimentally in Section 4.4.

In short, OBC attempts to capture the benefits of both experience replay (less feature-extractor overfitting), and training the classifier only with memory data (less prediction bias), in a best-of-both-worlds manner.

## 4 EXPERIMENTS

### 4.1 EXPERIMENTAL SETTINGS

**Datasets and Experimental Setup**  We use four datasets of varying difficulty. The FashionMNIST dataset (Xiao et al., 2017) contains 60,000 grayscale images of clothing items split in 10 classes. CIFAR-10 and CIFAR-100 (Krizhevsky, 2009) each contain 50,000 color images, with the ones in CIFAR-10 being divided in 10 classes, and the ones in CIFAR-100 in 100 classes. Finally, tinyImageNet (Le & Yang, 2015) contains 100,000 color images of 200 different classes. The tinyImageNet dataset is the most challenging dataset widely used in the task-free continual learning literature, mainly due to its large number of classes (200) and the small number of data instances per class (500). Our experimental setup closely follows previous work (Aljundi et al., 2019a; Jin et al., 2021; Caccia et al., 2022). We use class-incremental streams that are presented to the learner online, in small minibatches. For FashionMNIST and CIFAR-10, we use five binary tasks; for CIFAR-100 and tinyImageNet, we use ten and twenty tasks, respectively, each containing ten classes. We follow the majority of past work (Aljundi et al., 2019c; Chrysakis & Moens, 2020; Aljundi et al., 2019a) by not using data augmentation in our experiments.

**Methods**  Experience replay (ER) (Isele & Cosgun, 2018; Chaudhry et al., 2019) is the most fundamental continual learning baseline. It performs replay using a memory that is populated using reservoir sampling (Vitter, 1985). Memory-replay only (MRO) also uses reservoir sampling to populate the memory, but instead of using both stream and memory data, it trains the model using only data from the memory. Maximally-interfered retrieval (MIR) (Aljundi et al., 2019a) is an extension of ER that replays the instances which would experience the largest loss increases, if the model were to be updated using only the current minibatch of observations. Class-balancing reservoir sampling (CBRS) (Chrysakis & Moens, 2020) uses a memory population algorithm that attempts to maintain the memory as class-balanced as possible at all times. Greedy sampler and dumb learner (GDUMB) (Prabhu et al., 2020) also uses a class-balancing memory population algorithm and trains the model using only data stored in memory.[3] Gradient-based memory editing (GMED) (Jin et al., 2021) edits the data stored in memory in order to increase their loss values and make them more challenging. Finally, asymmetric cross entropy (ACE) (Caccia et al., 2022) modifies the traditional cross-entropy loss with class-masking, which reduces representation drift during continual learning.

**Architectures and Hyperparameters**  Similar to previous work (Lopez-Paz & Ranzato, 2017; Aljundi et al., 2019a), we use a reduced ResNet-18 architecture (He et al., 2016) for CIFAR-10, CIFAR-100, and tinyImageNet. For the simpler FashionMNIST, we use a simple convolutional neural network (CNN). For more information on these architectures, please refer to the appendix. Following previous work (Aljundi et al., 2019a; Jin et al., 2021), we use a learning rate of 0.1 when using the reduced ResNet-18 architecture. When using the simpler CNN, we use a learning rate of 0.03. The stream and replay batch sizes were both set to 10, in accordance with past work. The batch size of OBC was set to 50 (please refer to the appendix for a sensitivity analysis.) Past work typically

---

[3]We adapt GDUMB for use in task-free continual learning (in this setting, the learner should always be available for inference, while in (Prabhu et al., 2020), the model is trained only after the entire stream is observed).

Table 2: Comparison of various task-free continual learning methods with and without OBC, over four datasets. We report the accuracy after training (Acc.) and the information retention averaged over the stream (Av. IR). All entries are $95\%$-confidence intervals over 15 runs.

| | FashionMNIST | | CIFAR-10 | | CIFAR-100 | | tinyImageNet | |
| | Acc. | Av. IR | Acc. | Av. IR | Acc. | Av. IR | Acc. | Av. IR |
|---|---|---|---|---|---|---|---|---|
| ER | $83.4 \pm 0.5$ | $86.5 \pm 0.2$ | $45.5 \pm 1.7$ | $60.0 \pm 0.4$ | $19.3 \pm 0.7$ | $27.0 \pm 0.3$ | $13.5 \pm 0.5$ | $15.4 \pm 0.2$ |
| +OBC | $84.9 \pm 0.3$ | $89.8 \pm 0.1$ | $54.3 \pm 1.1$ | $67.5 \pm 0.4$ | $25.1 \pm 0.6$ | $35.2 \pm 0.4$ | $16.9 \pm 0.5$ | $21.7 \pm 0.2$ |
| MIR | $83.7 \pm 0.4$ | $87.5 \pm 0.2$ | $47.1 \pm 1.5$ | $59.1 \pm 0.7$ | $18.8 \pm 0.7$ | $27.4 \pm 0.4$ | $12.4 \pm 0.7$ | $15.8 \pm 0.2$ |
| +OBC | $84.9 \pm 0.3$ | $89.7 \pm 0.1$ | $53.4 \pm 1.3$ | $66.8 \pm 0.4$ | $23.8 \pm 0.6$ | $34.9 \pm 0.4$ | $15.5 \pm 0.6$ | $21.7 \pm 0.2$ |
| CBRS | $82.9 \pm 0.5$ | $84.8 \pm 0.2$ | $44.9 \pm 1.8$ | $57.6 \pm 0.4$ | $19.2 \pm 0.7$ | $25.7 \pm 0.3$ | $13.4 \pm 0.6$ | $14.8 \pm 0.2$ |
| +OBC | $84.5 \pm 0.3$ | $89.2 \pm 0.1$ | $53.4 \pm 1.1$ | $66.7 \pm 0.4$ | $25.4 \pm 0.6$ | $35.0 \pm 0.3$ | $17.0 \pm 0.5$ | $21.6 \pm 0.2$ |
| GMED | $83.9 \pm 0.7$ | $87.0 \pm 0.2$ | $46.2 \pm 1.8$ | $60.4 \pm 0.5$ | $19.8 \pm 0.8$ | $27.2 \pm 0.4$ | $13.7 \pm 0.7$ | $16.1 \pm 0.3$ |
| +OBC | $85.1 \pm 0.4$ | $90.1 \pm 0.3$ | $54.7 \pm 1.3$ | $67.9 \pm 0.5$ | $25.4 \pm 0.8$ | $35.4 \pm 0.4$ | $17.1 \pm 0.6$ | $21.9 \pm 0.3$ |

uses memory sizes in the range of $1\%$–$10\%$ of the size of the stream. Hence, unless explicitly mentioned otherwise, we set the memory size to 1000 for both FashionMNIST (approximately 2%) and CIFAR-10 (2%), and to 2500 for CIFAR-100 (5%) and 5000 for tinyImageNet (5%).[4] For OBC, and only when training the classifier, we use a batch size of 50, and a label-smoothing[5] factor of 0.5 (we perform a sensitivity analysis of these two hyperparameters in the appendix). Method-specific hyperparameters were set to the values given in their respective papers.

**Evaluation Metrics** We calculate the *accuracy* and the proposed *prediction bias* metric we proposed in Section 3.3, after the entire stream has been observed. Both of these metrics are calculated with respect to the unseen test set. These metrics inform us about how well the model has learned *after* the end of learning. We also use the *information retention* metric (accuracy computed with respect to past observations) proposed in Cai et al. (2021), averaged over the entire stream. This metric is a form of continual evaluation that evaluates how well each method performs, not only *after* the end of learning, but *during* its entire length. Continual evaluation is crucial for real-world applications where training and inference take place interchangeably.

## 4.2 APPLYING OBC

At first we apply OBC to four state-of-the-art, task-free continual learning methods. Namely, we compare the performance (final accuracy and information retention averaged over the stream) of ER, MIR, CBRS, and GMED, with and without OBC on four different datasets (see Table 2). (Due to of lack of space, we report the prediction bias numbers of these experiments in the appendix.) We observe that OBC improves both the learning performance of each method over the continuum (Av. IR), and their final accuracy (Acc.), for all four datasets. The learning benefit that OBC provides is especially prominent in CIFAR-10, CIFAR-100, and tinyImageNet. We note that methods that perform post-training bias correction (as the one we proposed in Section 3.4, or the NCM approach proposed by Mai et al. (2021)) would only affect the final accuracy of each method, but not the average information retention over the stream, because, by definition, the bias correction takes place after the end of training.

Since OBC only modifies the way the classifier is trained, we argue that when applying bias correction to various methods, we can essentially compare the quality of the representations that their feature extractor learns. In the results presented here, there is no clear best method in that respect, but future work could specifically focus on improving feature-extractor representation learning.

---

[4]To avoid potential confusion, we would like to point out that some previous works (e.g., Aljundi et al. (2019a)) do not report the size of the entire memory, but the memory size divided by the number of classes instead. Along similar lines, the memory sizes used here are 100, 100, 25, and 50, respectively.

[5]We found that when label smoothing is applied in combination with other methods (ER, GDUMB, ACE, etc.) it invariably leads to reduced performance, possibly because it is not applied specifically to the training of the classifier, but to the entire model.

Table 3: Comparison of various bias-correction methods for three memory sizes on CIFAR-10. We report the accuracy after training (Acc.), the information retention averaged over the stream (Av. IR), and the prediction bias (Pred. Bias) of each approach. All entries are $95\%$-confidence intervals over 15 runs.

|  | 500 | | | 1000 | | | 2500 | | |
| --- | --- | --- | --- | --- | --- | --- | --- | --- | --- |
|  | Acc. | Av. IR | Pred. Bias | Acc. | Av. IR | Pred. Bias | Acc. | Av. IR | Pred. Bias |
| ER | $37.4 \pm 1.4$ | $55.0 \pm 0.3$ | $28.9 \pm 2.4$ | $45.5 \pm 1.7$ | $60.0 \pm 0.4$ | $18.1 \pm 2.8$ | $56.1 \pm 1.8$ | $64.3 \pm 0.5$ | $8.3 \pm 1.7$ |
| GDUMB | $37.5 \pm 1.2$ | $54.5 \pm 0.3$ | $16.8 \pm 2.2$ | $42.2 \pm 1.7$ | $57.9 \pm 0.3$ | $10.8 \pm 2.7$ | $49.3 \pm 1.5$ | $61.3 \pm 0.4$ | $6.6 \pm 1.6$ |
| MRO | $34.4 \pm 1.3$ | $54.9 \pm 0.3$ | $9.2 \pm 3.3$ | $38.2 \pm 1.6$ | $58.1 \pm 0.3$ | $10.5 \pm 2.7$ | $46.0 \pm 1.5$ | $61.6 \pm 0.3$ | $7.7 \pm 1.7$ |
| ACE | $46.7 \pm 1.4$ | $63.3 \pm 0.4$ | $4.6 \pm 1.6$ | $51.6 \pm 1.9$ | $65.3 \pm 0.5$ | $5.3 \pm 2.2$ | $53.8 \pm 1.6$ | $67.5 \pm 0.4$ | $5.0 \pm 1.4$ |
| ER+OBC | $46.8 \pm 1.3$ | $63.4 \pm 0.4$ | $1.3 \pm 0.3$ | $54.3 \pm 1.0$ | $67.6 \pm 0.4$ | $0.9 \pm 0.2$ | $61.5 \pm 1.3$ | $71.0 \pm 0.4$ | $1.0 \pm 0.4$ |

Table 4: Comparison of various bias-correction methods for three memory sizes on CIFAR-100. We report the accuracy after training (Acc.), the information retention averaged over the stream (Av. IR), and the prediction bias (Pred. Bias) of each approach. All entries are $95\%$-confidence intervals over 15 runs.

|  | 1000 | | | 2500 | | | 5000 | | |
| --- | --- | --- | --- | --- | --- | --- | --- | --- | --- |
|  | Acc. | Av. IR | Pred. Bias | Acc. | Av. IR | Pred. Bias | Acc. | Av. IR | Pred. Bias |
| ER | $12.8 \pm 0.6$ | $23.4 \pm 0.3$ | $57.2 \pm 2.4$ | $19.3 \pm 0.7$ | $26.9 \pm 0.3$ | $34.7 \pm 2.0$ | $23.1 \pm 0.8$ | $27.8 \pm 0.4$ | $25.5 \pm 1.6$ |
| GDUMB | $11.2 \pm 0.5$ | $19.2 \pm 0.2$ | $26.2 \pm 2.6$ | $15.3 \pm 0.7$ | $21.5 \pm 0.3$ | $16.4 \pm 2.3$ | $18.6 \pm 0.6$ | $22.4 \pm 0.3$ | $10.2 \pm 1.2$ |
| MRO | $10.2 \pm 0.5$ | $19.0 \pm 0.2$ | $23.8 \pm 3.0$ | $14.7 \pm 0.6$ | $21.4 \pm 0.3$ | $17.0 \pm 2.2$ | $18.1 \pm 0.6$ | $22.4 \pm 0.3$ | $11.1 \pm 1.3$ |
| ACE | $17.8 \pm 0.5$ | $27.0 \pm 0.3$ | $8.7 \pm 1.2$ | $22.4 \pm 0.6$ | $29.3 \pm 0.4$ | $6.6 \pm 1.1$ | $25.7 \pm 0.7$ | $30.2 \pm 0.4$ | $4.8 \pm 0.5$ |
| ER+OBC | $19.0 \pm 0.6$ | $31.0 \pm 0.3$ | $2.5 \pm 0.5$ | $25.1 \pm 0.5$ | $35.2 \pm 0.4$ | $2.7 \pm 0.4$ | $30.7 \pm 0.5$ | $37.1 \pm 0.5$ | $1.8 \pm 0.3$ |

## 4.3 COMPARISON WITH OTHER BIAS CORRECTION APPROACHES

At this point, we will compare OBC to three other methods that perform implicit bias correction (in the sense that they were not designed specifically to correct for prediction bias, but they do so anyway). In particular, we compare OBC to GDUMB, MRO (they are not biased by the stream in the first place since they both train models using memory data only), and ACE (it uses both stream and replay data, but with a masked cross-entropy, which is aimed to reduce representation drift, but also does not lead to biased predictions). We also include ER in this comparison, as a biased baseline. The comparison takes place on CIFAR-10 (see Table 3) and CIFAR-100 (see Table 4). We use memory sizes that correspond to $1\%$, $2\%$, $5\%$ of the size of CIFAR-10, and $2\%$, $5\%$, $10\%$ of the size of CIFAR-100. For CIFAR-10, we can see that OBC outperforms GDUMB and MRO, and is competitive with ACE for very small memories of 500 instances. For the other two memory sizes, OBC outperforms GDUMB, MRO, and ACE, with respect to all three metrics used. In CIFAR-100, OBC outperforms GDUMB, MRO, and ACE, for all three memory sizes. Interestingly, we point out that for both CIFAR-10 and CIFAR-100, the gaps in accuracy and average information retention between OBC and the other three bias-correction methods increase with larger memory sizes.

Table 5: Comparison of OBC with and without (OBC – SC) a surrogate classifier over two datasets. We report the accuracy after training (Acc.), the information retention averaged over the stream (Av. IR), and the prediction bias (Pred. Bias) of each approach. All entries are $95\%$-confidence intervals over 15 runs.

|  | CIFAR-10 | | | CIFAR-100 | | |
| --- | --- | --- | --- | --- | --- | --- |
|  | Acc. | Av. IR | Pred. Bias | Acc. | Av. IR | Pred. Bias |
| ER | $45.5 \pm 1.7$ | $60.0 \pm 0.4$ | $18.1 \pm 2.8$ | $19.1 \pm 0.7$ | $27.0 \pm 0.3$ | $34.7 \pm 2.0$ |
| +OBC | $54.3 \pm 1.0$ | $67.6 \pm 0.3$ | $0.9 \pm 0.2$ | $25.1 \pm 0.5$ | $35.2 \pm 0.3$ | $2.7 \pm 0.4$ |
| +OBC – SC | $42.8 \pm 2.3$ | $58.3 \pm 0.4$ | $6.1 \pm 1.3$ | $21.0 \pm 0.6$ | $25.6 \pm 0.5$ | $6.1 \pm 0.5$ |

### 4.4 THE NEED FOR A SURROGATE CLASSIFIER

Finally, we experimentally motivate the introduction of a surrogate classifier in the OBC paradigm. We compare the performance of OBC when applied to ER, with and without the use of a surrogate classifier. In the latter case, the computational graph in the left part of Figure 1 flows from the feature extractor to the classifier, but the classifier is not updated with the resulting gradients. The right part of Figure 1 remains unchanged.

The results are presented in Table 5. We observe that both for CIFAR-10 and CIFAR-100, the alternative formulation of OBC without a surrogate classifier (OBC – SC) performs worse than OBC with a surrogate classifier. These results validate the use of the surrogate classifier. Since OBC is applied only for the optimization of the classifier in both cases, we can only conclude that not using a surrogate classifier negatively affects the feature extractor. In other words, it appears that the feature extractor learns better representations when it is trained in combination with the biased surrogate classifier than when it is trained combined with the unbiased classifier. Future work focusing on better representation learning in task-free settings could further interpret this finding.

### 4.5 DISCUSSION

As we saw in Section 4.2, OBC performs reliably for multiple methods and over multiple datasets. In Section 4.3, we showed that it also outperforms other methods that perform bias correction implicitly, for different memory sizes, and for two different datasets.

In our view, one important point this work makes is that, in the context of task-free continual learning, we should not necessarily be viewing a neural network as a single black box. Instead, we should make a distinction between the feature extractor and the classifier, because of their different learning capacities. In particular, our OBC approach trains the classifier of the neural network online using only memory data, in order to mitigate its prediction bias. Moreover, the feature extractor and the surrogate classifier are trained via experience replay, in order to prevent the overfitting that would result if the feature extractor were to be trained only using memory data. An interesting direction for future work would be to investigate the extent to which the data-sampling bias affects the feature extractor of the learner.

One potential limitation of OBC is that it does not improve the feature extractor of the model. Consequently, the final performance after applying OBC will depend on the quality of the representations learned by the method being wrapped. Therefore, another interesting avenue for future work, would be to focus on how to improve task-free continual representation learning.

Finally, we hypothesize that OBC can be used to disentangle the confounding factors of prediction bias and representation learning. In particular, when two task-free continual learning methods use the same memory population algorithm and their classifiers are both trained using OBC, any differences in learning performance can be explained by the quality of the representations learned by their respective feature extractors. For example, if a method A learns better representations but is also more biased than approach B, it is possible that method B would achieve higher accuracy. But, if we correct for the bias in both methods using our proposed bias-correction approach, we expect method A to outperform method B, since A learns better representations than B.

## 5 CONCLUSION

To summarize, this work discusses the issue of prediction bias in task-free continual learning. In particular, we provided a concrete explanation of how this bias is caused—namely, by the experience-replay paradigm favoring current stream observations over the past data stored in memory. Subsequently, we proposed an evaluation metric that quantifies the prediction bias of a model on an unseen test set. More importantly, we proposed a simple approach called Online Bias Correction (OBC) that can correct this prediction bias online throughout the entire duration of learning. Because of its online nature, OBC is especially useful for real-world applications in which a model needs to learn and perform inference at the same time. Also, OBC is trivial to implement and can be applied as a wrapper over other task-free continual learning methods in order to increase their accuracy and their information retention, and reduce their prediction bias.

ACKNOWLEDGEMENTS

This work is part of the CALCULUS project (https://calculus-project.eu/). The CALCULUS project is funded by the ERC Advanced Grant H2020-ERC-2017-ADG 788506.

REPRODUCIBILITY

Our code can be found at https://github.com/chrysakis/OBC.

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

## A  HYPERPARAMETER SENSITIVITY

We now examine how sensitive OBC is to its hyperparameters. (Note that other hyperparameters such as the stream batch size are applicable, not directly to OBC, but only to the method that OBC wraps.) In Figure 2, we present the final accuracy of ER+OBC for various values of the OBC batch size (10, 20, 50, 100, 200), and the label-smoothing factor (0.2,...,0.8).[6] The accuracies correspond to CIFAR-100 for memory sizes of 1000 and 2500, and are presented as 95%-confidence intervals. The batch-size curves are very similar for both memory sizes. Higher batch sizes are correlated with higher accuracies, but with diminishing returns. For the label-smoothing factor, we can see that relatively large values (0.7 or 0.8) lead to lower accuracies for both memory sizes. In addition, only for the smaller memory of 1000 instances, we also observe that a relatively small label-smoothing factor of 0.2 or 0.3 also leads to slightly reduced accuracies.

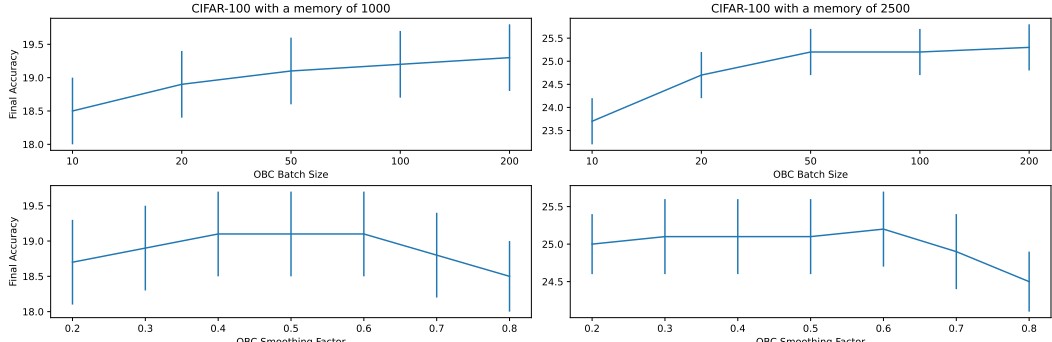

Figure 2: We examine the sensitivity of OBC with respect to its two hyperparameters, namely, the label-smoothing factor, and the batch size. The comparison is performed on CIFAR-100 for two memory sizes. The results are presented as 95%-confidence intervals over 15 runs.

Table 6: (left) A simple convolutional block; (middle) The Convolutional Neural Network (CNN) architecture used in the FashionMNIST experiments. (right) The reduced ResNet-18 architecture used for CIFAR-10, CIFAR-100, and tinyImageNet, is built using the BasicBlock($n_f, n_b, n_s$) from (He et al., 2016), where $n_f$ is the number of convolutional filters, $n_b$ is the number of sub-blocks per block, and $n_s$ is the stride of the layer.

| ConvBlock | CNN | Reduced ResNet-18 |
|---|---|---|
| Conv2D($n_{in}, n_{out}$) | ConvBlock($1, 32$) | BasicBlock($20, 2, 1$) |
| ReLU | ConvBlock($32, 64$) | BasicBlock($40, 2, 2$) |
| BatchNorm2D($n_{out}$) | Linear($64, c$) | BasicBlock($80, 2, 2$) |
| Conv2D($n_{out}, n_{out}$) | | BasicBlock($160, 2, 1$) |
| ReLU | | AveragePooling |
| BatchNorm2D($n_{out}$) | | Linear($160, c$) |
| MaxPooling2D($2, 2$) | | |

## B  ARCHITECTURES

In Table 6, we report the architectures used in our experiments.

## C  RESULTS ON CORE50

In Table 7, we present results on the CORe50 dataset (Lomonaco & Maltoni, 2017). This dataset contains sequential video frames of 50 objects filmed during 11 different sessions, with each session

---

[6]We remind the reader, that for all previous experiments we use an OBC batch size of 50 and a label-smoothing factor of 0.5.

Table 7: Comparison of four methods with and without OBC using the CORe50 dataset. All entries are 95%-confidence intervals over 15 runs.

|        | Acc.          | Av. IR        | Pred. Bias    |
|--------|---------------|---------------|---------------|
| ER     | $26.8 \pm 1.2$ | $76.9 \pm 0.7$ | $24.8 \pm 1.7$ |
| +OBC   | $30.0 \pm 1.2$ | $83.0 \pm 0.5$ | $2.2 \pm 0.5$ |
| MIR    | $27.8 \pm 1.6$ | $80.1 \pm 0.7$ | $25.3 \pm 3.6$ |
| +OBC   | $30.8 \pm 0.9$ | $84.2 \pm 0.6$ | $2.1 \pm 0.4$ |
| CBRS   | $27.1 \pm 0.2$ | $75.3 \pm 0.7$ | $24.0 \pm 1.7$ |
| +OBC   | $30.2 \pm 0.1$ | $82.2 \pm 0.6$ | $2.3 \pm 0.5$ |
| GMED   | $26.9 \pm 1.3$ | $77.1 \pm 0.9$ | $24.3 \pm 1.7$ |
| +OBC   | $30.2 \pm 1.2$ | $83.0 \pm 0.5$ | $2.3 \pm 0.5$ |

Table 8: Comparison of the prediction bias of four task-free continual learning methods with and without OBC, over four datasets. The prediction bias is always calculated after the end of the stream, with respect to the unseen test set. All entries are 95%-confidence intervals over 15 runs.

|        | FashionMNIST  | CIFAR-10       | CIFAR-100     | tinyImageNet  |
|--------|---------------|----------------|---------------|---------------|
| ER     | $0.9 \pm 0.2$ | $17.0 \pm 2.4$ | $33.8 \pm 2.0$ | $25.3 \pm 1.7$ |
| +OBC   | $0.2 \pm 0.1$ | $0.9 \pm 0.3$  | $2.7 \pm 0.5$ | $2.4 \pm 0.5$ |
| MIR    | $0.9 \pm 0.2$ | $16.0 \pm 2.1$ | $35.8 \pm 2.0$ | $28.3 \pm 2.1$ |
| +OBC   | $0.2 \pm 0.1$ | $1.0 \pm 0.5$  | $3.2 \pm 0.5$ | $3.5 \pm 0.7$ |
| CBRS   | $1.0 \pm 0.2$ | $18.7 \pm 2.6$ | $33.7 \pm 2.1$ | $25.3 \pm 1.9$ |
| +OBC   | $0.2 \pm 0.1$ | $1.0 \pm 0.3$  | $2.5 \pm 0.4$ | $2.4 \pm 0.4$ |
| GMED   | $0.9 \pm 0.3$ | $16.9 \pm 2.6$ | $33.6 \pm 2.0$ | $25.1 \pm 1.7$ |
| +OBC   | $0.2 \pm 0.1$ | $1.0 \pm 0.5$  | $2.6 \pm 0.5$ | $2.3 \pm 0.6$ |

consisting of approximately 300 frames. As Lomonaco & Maltoni (2017) suggest, sessions 3, 7, and 10 are used for evaluation purposes (approximately 45,000 images), and the remaining 8 sessions are used to construct the stream (approximately 120,000 images). The stream consists of ten tasks of five objects each. We used the reduced ResNet-18 architecture described earlier, with a learning rate of 0.01 and a memory size of 2400 (2% of the size of the stream). The remaining hyperparameters are identical to the ones described in the main paper. As before, we compare four task-free continual learning methods, with and without OBC, and we report the accuracy and the prediction bias calculated with respect to the unseen test set, and the information retention (accuracy on past observations) averaged over the stream. We observe that the use of OBC results to higher accuracy and average information retention, and lower prediction bias, for all four methods.

## D    REDUCTIONS IN PREDICTION BIAS WHEN USING OBC

In Table 8, we report the prediction bias results from the experiments of Section 4.2, which were not included in the main text due to lack of space. These results suggest that the use of OBC leads to significant reductions in prediction bias. These reductions are consistent for all four methods and all four datasets.

## E    A NUMERICAL EXAMPLE OF DATA-SAMPLING BIAS

Consider a learner that has previously observed data 990 instances, and these instances are currently stored in the learner's memory. A new minibatch of 10 instances is now observed by the learner. If the learner wants to update the model in an unbiased way, all $990 + 10$ data instances should be equally likely to participate in this update. Assuming that 20 instances will be used in the update, the probability that any instance participates is therefore $20/(990 + 10) = 0.02$. In experience

replay, however, the 10 instances of the new minibatch are guaranteed to participate in the update (therefore with a probability of 1), while an equal number of instances would be randomly sampled from memory. Hence, each memory instance is sampled with a probability of $10/990 \simeq 0.01$. Therefore, in experience replay, the probability that a new instance will participate in the update is $1/0.01 = 100$ times larger than that of a random memory instance, while in the unbiased case, all $990 + 10$ are equally likely to be sampled, regardless of whether they are new observations or stored in memory.

## F  CONTINUAL LEARNING, ONLINE LEARNING, AND DOMAIN ADAPTATION

For the sake of clarity we provide a high-level comparison of continual learning, online learning, and domain adaptation. In continual learning, the data distribution changes over time and, in general, previous work assumes that all observed instances are equally important, and that there is no distributional mismatch between the training and testing data (Jin et al., 2021). *Online learning* is similar to continual learning in the sense that the data distribution changes over time. Yet, the goal in online learning is for the model to appropriately adapt to the current state of the data distribution (Fontenla-Romero et al., 2013). Therefore, currently observed instances are in essence more important than previous ones. Finally, in the problem of *domain adaptation*, a model is trained offline using a set of training data. Subsequently, the model needs to perform inference on data generated by a data distribution that is different (to a degree) from the one that generated the training data (Redko et al., 2019).

