# OpenReview forum: "Online Bias Correction for Task-Free Continual Learning"
_ICLR.cc/2023/Conference — ICLR 2023 poster_

### Official Review · Reviewer_cbbB · 2022-10-23

**Confidence:** 3
**Correctness:** 3
**Technical Novelty And Significance:** 3
**Empirical Novelty And Significance:** 3
**Recommendation:** 6

**Clarity, Quality, Novelty And Reproducibility:**

The paper is easy to read, but the technical part can be improved. At least, I would like to see the problem formally defined especially around how the data are generated and how the learned model is evaluated. As I mentioned, the setups of the experiments are not clear to me, and it may be difficult to reproduce them.

**Strength And Weaknesses:**

# Strengths
- The paper points out an important weakness of experience replay that seems relevant to the research of continual learning.
- The proposed method is simple and widely applicable without big implementation or computational overhead.
- The experiments consistently show its effectiveness.

# Weaknesses
- The problem setup on the mathematical level is not clear to me. The paper says that it does not assume strong assumptions, but without any assumptions on how the distribution can change, I don't think we can say much about the learnability. One possible direction may be considering some form of cumulative regret as in online learning, but the paper does not seem to be considering such a setup either.
- The setups of the experiments are also not clear to me. How the samples are drawn in the stream from the dataset (e.g., gradually including new classes) seems important, but I could not find any part mentioning that.
- I could not understand how the proposed method makes the right amount of bias correction. If the effect is too strong, it can be biased in the opposite way, to the past experience side.
- The measure of the "bias" is misleading because it does not quantify the bias in the sense of the standard statistics. (The expectation is not taken over all possible draws of the training samples.) It is also a quite weak measure because it averages over the samples.

**Summary Of The Paper:**

This paper proposes a method for correcting bias of continual learning method using experience replay. The proposed method uses a memory from past examples to train the last layer of a neural network trained by an experience-replay-based method. This prevents the model from being biased to recently observed samples. By fixing layers other than the last, it also prevents overfitting to the chosen samples from the memory. The authors show that the proposed method indeed reduces the bias and improves the overall performance.

**Summary Of The Review:**

Although the work is well-motivated, the paper is easy to read, and the experiments show the effectiveness of the proposed method, some important details are not clear, and the theoretical part might have a weakness. Also, there is no evidence that the proposed method really corrects the bias theoretically or even empirically because the measure of the bias does not seem to capture the bias that the authors are trying to mitigate. I vote for weak reject.

---

> ### Author Response · Authors · 2022-11-10
> **Reply to Reviewer cbbB**
>
> #### **Learnability of Task-Free Continual Learning**
>
> Unfortunately, the task-free continual learning problem is severely underexplored, and, to the best of our knowledge, there is no work formally examining whether the problem is provably learnable for all possible nonstationary data distributions. This question is undoubtedly very important, but we think that answering it is outside the scope of the present work.
>
>
> &nbsp;
> #### **Regret**
>
> To the best of our knowledge, regret is not a suitable measure to evaluate continual learning performance. Regret quantifies how good the model's predictions are with respect to the present observations, but in continual learning, the goal is not only to learn the present observations but also to keep on remembering past observations. The average information retention metric that we have used attempts to quantify exactly that ability over time, namely, to learn the present and keep remembering the past.
>
>
> &nbsp;
> #### **Experimental Setup**
>
> Our experimental setup closely follows previous work (e.g., [1, 2]).  In particular, we use class-incremental continuums that are presented to the model in an online manner (in small mini-batches of 10 instances). (In a class-incremental continuum with $c_t$ classes per task, the model is first presented with the first $c_t$ classes mixed together, then with the following $c_t$ classes mixed together, and so on, until all classes have been observed). For FashionMNIST and CIFAR-10, we used five binary tasks, for CIFAR-100 ten tasks of ten classes each, and for tinyImageNet, twenty tasks of ten classes each. We will edit our submission to better clarify this point. Crucially, none of the methods we use in our experiments assumes anything, or is informed in any way, about the structure of the stream.
>
> [1] Jin, X., Sadhu, A., Du, J., & Ren, X. (2021). Gradient-based Editing of Memory Examples for Online Task-free Continual Learning. NeurIPS.
>
> [2] Aljundi R., Caccia L., Belilovsky E., Caccia M., Lin M., Charlin L., and Tuytelaars T. (2019). Online continual learning with maximally interfered retrieval. Proceedings of the 33rd International Conference on Neural Information Processing Systems.
>
>
> &nbsp;
> #### **Right Amount of Bias Correction**
>
> Note that bias correction takes place implicitly (in the sense that we do not add or subtract a bias correction term to an estimator to make it unbiased), and it is dictated by the contents of the memory. Since the training step of the classifier (final layer of the model) takes place after the current observations have been processed by the memory population algorithm (see Algorithm 1), the memory is essentially a summary of all the instances the model has observed. Therefore, the classifier is only trained using data from this summary, so that it is not biased towards the present observations. Moreover, if the memory population algorithm is reservoir sampling [3], as is commonly the case, there is a theoretical guarantee that the memory contains at all times an unbiased subset of all past observations.
>
> [3] Vitter, J. S. (1985). Random sampling with a reservoir. ACM Transactions on Mathematical Software (TOMS), 11(1), 37-57.
>
>
> &nbsp;
> #### **Prediction Bias**
>
> We agree that the term bias could possibly be confusing without context. That is why we refer to it as prediction bias and have given it a precise definition within the text.
>
> We did not understand what you meant by "weak measure." We use prediction bias as an evaluation metric, and we compute it with respect to all instances of an unseen test set (similarly to how the accuracy is computed). Could you elaborate further on this point?
>
> We respectfully disagree that there is no evidence that we mitigate prediction bias. In Tables 3 and 4, there are very large reductions in the prediction bias of ER when OBC is used, and in Table 5 we show that when using a surrogate classifier the prediction bias is much better mitigated, compared to when not using a surrogate classifier. In other words, we have showed all components of our proposed approach contribute to the reduction of prediction bias (and also to the increase of accuracy and average information retention).
>
>
> &nbsp;
> ***
> We want to thank you for you review. Please let us know if any of your points were not addressed properly, or if you have any additional questions.

---

> > ### Comment · Reviewer_cbbB · 2022-11-20
> > **About the setup and the bias**
> >
> > I thank the authors for their answers.
> >
> > ## Learnability of Task-Free Continual Learning
> > I now understood that the regret won't be a right metric.
> >
> > But I don't think I have understood the problem setup even at an informal level (before talking about a formal definition). The paper explains, "continual learning is the process of incrementally aggregating knowledge from data that are generated by a non-stationary distribution," but what kind of non-stationarity are we considering? Let's say we want to learn a binary classifier, but the optimal classification rule changes so much over time that the optimal decisions at one point of time and another point contradict each other for every instance. What function should we maintain as our "aggregated" hypothesis in this case?
> >
> > ## Bias
> > In Section 3.3, the paper defines the bias as the discrepancy of $E[y_i]$ and $E[h(x_i; \theta_h)]$. This is weak because it only compares the outputs averaged over the instances. That cannot capture the difference on the functional level.

---

> > > ### Author Response · Authors · 2022-11-23
> > > **Reply to Additional Questions**
> > >
> > > Dear reviewer,
> > >
> > > thank you for participating in the discussion. Please find our reply below:
> > >
> > > &nbsp;
> > > #### **Problem Setup**
> > >
> > > Previous work assumes that there exists an optimal classification rule over all data that the model has observed, and the goal is to approximate this rule. In other words, the goal is to learn a model as if all past observations were available at the same time. The difficulty of this setting is not caused by changes in $p(y | \boldsymbol{x})$, but by the fact that the instances $\boldsymbol{x}$ that the learner observes are presented in small minibatches that are not sampled iid from the entire pool of data (e.g., the model might first observe all instances of class $0$ and then all instances of class $1$).
> > >
> > > If we understood correctly the example that you described, during the stream, each instance $\boldsymbol{x}_i$ appears once with the label $0$ and once with the label $1$. In this case, there is no optimal rule over the entire stream, hence the optimal "aggregated" hypothesis will be uninformative (e.g., for any instance $\boldsymbol{x}_i$, it will predict that labels $0$ and $1$ are equally likely).
> > >
> > >
> > > &nbsp;
> > > #### **Prediction Bias**
> > >
> > > Since the true data distribution is unknown, we do not think it is possible to capture the difference on a functional level. Therefore, we argue that the best we can do is to use a metric that averages over samples from the true data distribution (as the metric we proposed). Note that this also true for the accuracy metric, which is arguably the most widely used evaluation metric in the field of neural networks.
> > >
> > >
> > > &nbsp;
> > > ***
> > > We would be happy to discuss further in case you have more questions.

---

> > > > ### Comment · Reviewer_cbbB · 2022-11-25
> > > > **Additional comments**
> > > >
> > > > > Previous work assumes that there exists an optimal classification rule over all data that the model has observed, and the goal is to approximate this rule. In other words, the goal is to learn a model as if all past observations were available at the same time.
> > > >
> > > > I understand this setup to some extent. Is this the setup that the paper is working on too?
> > > > Also, could the authors tell me which previous work they are talking about?
> > > >
> > > >
> > > > > If we understood correctly the example that you described, during the stream, each instance $x_i$ appears once with the label 0 and once with the label 1.
> > > >
> > > > That is more or less what I was considering although I did not mean the conflicting labels occur because of stochasticity, but I meant they disagree because of the nonstationarity of the distribution.
> > > >
> > > >
> > > > > In this case, there is no optimal rule over the entire stream, hence the optimal "aggregated" hypothesis will be uninformative (e.g., for any instance $x_i$, it will predict that labels and are equally likely).
> > > >
> > > > It is fine that the optimal rule is useless when the task is too hard. The optimal rule can be useless even for the ordinary classification setup. What I care about is how to define the optimal rule. Saying that there is no optimal rule is not satisfactory because it means that the learning target cannot be well-defined.
> > > >
> > > >
> > > > > Since the true data distribution is unknown, we do not think it is possible to capture the difference on a functional level. Therefore, we argue that the best we can do is to use a metric that averages over samples from the true data distribution (as the metric we proposed).
> > > >
> > > > I cannot agree. The true distribution is unknown, but data following it are available, so the distribution is identifiable by estimation. The averages are rough summaries, and I do not think it is the best. (The authors would have to prove it if they claim it is the best.)
> > > > I am not against using the average if it is enough to show existing methods have bias. But we should not claim that there is no bias just because there is no difference in terms of the average.

---

> > > > > ### Author Response · Authors · 2022-11-27
> > > > > **Reply on Additional Comments**
> > > > >
> > > > > #### **Problem Setup**
> > > > >
> > > > > Indeed, we follow exactly this setup (please refer to e.g., [1, 2, 3, 4]). Please let us know if you have any further questions on the setup.
> > > > >
> > > > >
> > > > > &nbsp;
> > > > > #### **Optimal Rule in Hypothetical Example**
> > > > > Regarding the hypothetical example that you gave, by "there is no optimal rule" we meant that there is no optimal rule that is better than a random prediction. We agree that some problems can be so difficult that even knowing the optimal classification rule would not be of use.
> > > > >
> > > > >
> > > > > &nbsp;
> > > > > #### **Prediction Bias**
> > > > >
> > > > > To make sure we are on the same page, we note that by *prediction bias* we mean how much the true probability distribution over classes, $p(y)$, differs from the expected (with respect to the true data distribution) model output (i.e., predicted class probabilities). Within the paper, these two terms are called $\boldsymbol{p}$ and $\boldsymbol{q}$, respectively. Could you please let us know whether you disagree with the estimation of $\boldsymbol{p}$ or $\boldsymbol{q}$ (or both)?
> > > > >
> > > > > Please note that our proposed metric does not attempt to capture misclassification of individual instances, but systematic differences between $p(y)$ and the model predictions. To emphasize this point, consider a binary classification problem with $p(y=0)=0.6$ and $p(y=1)=0.4$. Consider also a classifier A that is optimal with respect to this problem, and a naive classifier B that randomly (i.e., regardless of the input $\boldsymbol{x}$) predicts 60% of the time class $0$, and 40% of the time class $1$. The two classifiers might differ a lot in terms of accuracy, but both of them are unbiased. The fact that classifier B makes a lot of classification errors does not necessarily mean that it has a systematic prediction bias.
> > > > >
> > > > >
> > > > > &nbsp;
> > > > > ***
> > > > > #### **References**
> > > > >
> > > > > [1] Jin, X., Sadhu, A., Du, J., and Ren, X. (2021). Gradient-based Editing of Memory Examples for Online Task-free Continual Learning. NeurIPS.
> > > > >
> > > > > [2] Aljundi, R., Kelchtermans, K., and Tuytelaars, T. (2019). Task-free continual learning. CVPR.
> > > > >
> > > > > [3] Aljundi R., Caccia L., Belilovsky E., Caccia M., Lin M., Charlin L., and Tuytelaars T. (2019). Online continual learning with maximally interfered retrieval. NeurIPS.
> > > > >
> > > > > [4] Caccia, L., Aljundi, R., Asadi, N., Tuytelaars, T., Pineau, J., and Belilovsky, E. (2021, September). New Insights on Reducing Abrupt Representation Change in Online Continual Learning. ICLR.

---

> > > > > > ### Comment · Reviewer_cbbB · 2022-12-08
> > > > > > **Suggestion**
> > > > > >
> > > > > > After the discussions with the authors and the reviewers, I conclude that my concerns boil down to the question on how effective the proposed bias correction is. I still think that the proposed metric of bias is too weak to measure the bias of a classifier, and the current experiments comparing the proposed bias metric might not be satisfactory enough to show the effectiveness.
> > > > > >
> > > > > > The reason why I think the proposed metric of bias (Eq. (2)) is weak is that it only measures the bias in p(y), but we need to correct that in terms of p(y | x).
> > > > > >
> > > > > > I suggest that the authors share any experiments showing that (i) previous methods are biased to recently observed samples, (ii) the proposed method corrects the bias, but (iii) it does not over-correct to the memory sample side.

---

> > > > > > > ### Author Response · Authors · 2022-12-10
> > > > > > > **Reply to Reviewer's Suggestion**
> > > > > > >
> > > > > > > Dear reviewer,
> > > > > > >
> > > > > > > thank you for your suggestions. Here is our reply.
> > > > > > >
> > > > > > >
> > > > > > > &nbsp;
> > > > > > > #### **Prediction Bias Metric**
> > > > > > >
> > > > > > > We want to note that $\boldsymbol{p} \simeq \mathbb{E}[p(y|x)]$ and $\boldsymbol{q} \simeq \mathbb{E}[h(y|x)]$, where $h$ is the predicted class probabilities of the model, and the expectations are with respect to the true data distribution. (We use the symbol $\simeq$ because we compute $\boldsymbol{p}$ and $\boldsymbol{q}$ as Monte-Carlo estimates of the true expectations, that is to say, by averaging the quantities within the expectations over samples drawn from the true data distribution, that is, the unseen test set.)
> > > > > > >
> > > > > > > Moreover, we want to clarify why we did not consider an alternative prediction bias metric that first calculates divergences between model predictions on individual test-set instances and their respective ground truth, and then averages over these divergences. This metric would not be suitable for quantifying prediction bias because it would take a small value only for high-accuracy unbiased classifiers but not for low-accuracy unbiased classifiers (please refer to our example about an optimal classifier and an unbiased classifier that outputs random labels, in our previous comment). Hence, this metric could be used to quantify classification performance, but not prediction bias.
> > > > > > >
> > > > > > >
> > > > > > > &nbsp;
> > > > > > > #### **Bias of ER-Based Methods**
> > > > > > >
> > > > > > > This phenomenon has been observed in previous works (e.g., [1, 2]), and in this paper, we have provided an explanation for it (Section 3.2 and Section E of the appendix), and proposed an approach to mitigate the bias online (which also leads to substantial increases in accuracy and information retention).
> > > > > > >
> > > > > > >
> > > > > > > &nbsp;
> > > > > > > #### **Bias Correction Results**
> > > > > > >
> > > > > > > We quickly provide the results of a single CIFAR-10 experiment. The test set contains $10,000$ instances ($1,000$ instances for each of the $10$ classes). The average (over the test set) output of a model trained with **ER** is
> > > > > > >
> > > > > > > $\boldsymbol{q}_1 = [0.04, 0.05, 0.06, 0.02, 0.03, 0.09, 0.13, 0.04, 0.28, 0.27]$
> > > > > > >
> > > > > > > and the average (over the test set) output of a model trained with **ER-OBC** is
> > > > > > >
> > > > > > > $\boldsymbol{q}_2 = [0.10, 0.09, 0.10, 0.09, 0.11, 0.10, 0.11, 0.11, 0.10, 0.09]$
> > > > > > >
> > > > > > > During the final part of the stream, both models observed instances of the last two classes (corresponding to the two rightmost elements of $\boldsymbol{q}_1$ and $\boldsymbol{q}_2$). We see that the average output of the ER model ($\boldsymbol{q}_1$) is clearly biased towards the last two classes, while that is not the case with the model trained with OBC ($\boldsymbol{q}_2$). This pattern is consistently present over all four datasets, and we will be happy to provide a visualization of these probabilities over multiple runs in the final version of the paper.
> > > > > > >
> > > > > > >
> > > > > > > &nbsp;
> > > > > > > #### **Bias Over-Correction**
> > > > > > >
> > > > > > > The reservoir-sampling algorithm, which is traditionally used to populate the memory, maintains at all times an iid subset of all observed data in memory [3, 4]. Therefore, this memory population algorithm does not favor past data over present data, nor the opposite. Hence, we are not sure how an over-correction using memory data could be possible. Instead, the crucial issue when training the model only using memory data is overfitting (because the memory is much smaller than the entire stream). Nonetheless, we have sidestepped that issue by using the data in memory to optimize only the final linear layer (and not the entire model).
> > > > > > >
> > > > > > >
> > > > > > > &nbsp;
> > > > > > > #### **References**
> > > > > > >
> > > > > > > [1] Pietro Buzzega, Matteo Boschini, Angelo Porrello, and Simone Calderara. Rethinking experience replay: a bag of tricks for continual learning. ICPR.
> > > > > > >
> > > > > > > [2] Zheda Mai, Ruiwen Li, Hyunwoo Kim, and Scott Sanner. Supervised contrastive replay: Revisiting the nearest class mean classifier in online class-incremental continual learning.
> > > > > > >
> > > > > > > [3] Vitter, J. S. (1985). Random sampling with a reservoir. ACM Transactions on Mathematical Software (TOMS), 11(1), 37-57.
> > > > > > >
> > > > > > > [4] Chaudhry, A., Rohrbach, M., Elhoseiny, M., Ajanthan, T., Dokania, P. K., Torr, P. H., and Ranzato, M. A. (2019). On Tiny Episodic Memories in Continual Learning. In Workshop on Multi-Task and Lifelong Reinforcement Learning, 2019.

---

### Official Review · Reviewer_yCcF · 2022-10-24

**Confidence:** 4
**Correctness:** 3
**Technical Novelty And Significance:** 2
**Empirical Novelty And Significance:** 1
**Recommendation:** 6

**Clarity, Quality, Novelty And Reproducibility:**

There are many points, where the paper is not sufficiently clear.
- It is not clear, how the task-free setup is instantiated in the theoretical discussion and in the experiments. It argues that in the task-free setup, no strong assumptions are made about the non-stationarity over the stream (neither task nor class incremental). What is the source of the unseen data here? Some future data coming after the last observation in the current stream? Or are these some examples generated in parallel with the whole stream (and therefore tracing the entire history of the distribution shifts)? Are the classes assumed to be stable or can they possibly change as well? Can there be more/less classes, Can these change in nature? Similarly, how is the experimental data organized for the continual setup with distribution shifts? Are these organized into binary or 10-wise tasks (as common in the task and or class incremental setup) without indicating this to the model? Or is the data randomly mixed? Where, how is the distribution shift happening? How is the test set constructed?
- In section 2.2 you claim that the larger the stream, the larger the memory will be. Why "will" it be? Or do you mean to say it "shall" be larger? You further claim to investigate algorithms with computation cost O(n) independent of m. Isn't the memory itself a function of n? Why would the cost be ever O(mn)?
- In the experimental section, accuracy, bias and information retention are reported. Is accuracy and bias calculated over test set? How is the test set constructed with respect to the shifting distribution in the stream? Is the av IR equivalent to accuracy over the train set (= the whole stream)? If not, how is it different?

I believe the ideas are also not particularly novel
- the hierarchical view of NNs as a feature extractor with a follow up classifier has been introduced previously for example in the works of Naftali Thisby and his group.
- So was the need to weight unequally the current and replay observations to compensate for the oversampling of current distribution (e.g. Ramapuram et al., Lifelong generative modeling, 2020)

There are some further clarity or quality issues
- in section 3.3. the prediction bias quantification is outlined. The prior class probabilities are defined as expectations over the one-hot class encodings. The expectation is taken with respect to what distribution. The notation suggests that the output p is again a class-long vector, is that so? The same goes for q as expectation of the classification outputs. What is the distribution used in this expectation?
- JS is suggested as symmetric divergence. Why the need for JS? What would be wrong with KL and using the prior for the weighting?
- section 4.2 you claim "we can essentially compare the quality of the representations that their feature extractor learns". How can you compare these?
- section 4.4. "the computational graph in the left part of Figure 1 flows from the feature extractor to the classifier, but the classifier is not updated with the resulting gradients". What do you mean? That in this setup you do not train the classifier? Or did you mean to say the feature extractor?

Reproducibility is limited:
- Authors do not indicate, if the code is anywhere available
- How are the expectations in 3.3 calculated?

More general question - you seem to be training the classifier and the surrogate classifier in a completely disconnected manner (no sharing of parameters or parameter passing, etc.). Isn't this eventually going to lead into a complete divergence between the two classifier models with which the feature extractor is unlikely to be able to cope?


**Strength And Weaknesses:**

Strength:
The paper looks into the very important problem of continual learning. It is very readable and easy to follow.

Weaknesses:
The paper lacks on clarity and quality at several major points. These are concretely listed in the next section. It is also of limited novelty - also clarified in the following section.

**Summary Of The Paper:**

The paper addresses the problem of recency bias in the task-free continual learning setup. It proposes a new metric to quantify prediction bias and a method to mitigate it through adjusting parameters of the final model layer post training. The authors then propose a online procedure for training and unbiasing the model over the data stream via an experience replay mechanism. They test the performance of the algorithm over multiple datasets comparing favourable to several baselines.

**Summary Of The Review:**

The topic of the paper - continual learning - is an important one and novel methods in this area would be of great interest to the community. However, I find the current paper lacking significantly in clarity, quality and novelty to be recommendable for publication at ICLR. I therefore recommend rejection from the conference.

---

> ### Author Response · Authors · 2022-11-10
> **Reply to Reviewer yCcF (Part 2/2)**
>
> #### **Surrogate Classifier Ablation**
>
> Here, we attempt to apply our OBC approach without a surrogate classifier. This process takes place in two parts as with regular OBC (see Figure 1 of the paper). The difference is that when the feature extractor is trained using both present and replay data (Figure 1, left) we do not use a surrogate classifier, but we compute the output of the model using the (main) classifier. However, in the backward pass that follows, only the weights of the feature extractor are updated. The right part of Figure 1 remains unchanged (i.e., the classifer is then trained using replay data only, while the feature extractor is frozen). Our results suggest that using a surrogate classifier performs much better than not using one.
>
>
> &nbsp;
> #### **Reproducibility Concerns**
>
> As we stated in our reproducibility statement, we will release our code (and all necessary config files) upon acceptance, to ensure that our work is reproducible. Regarding the expectations in the calculation of prediction bias, we reiterate that they are computed over the data in the unseen test set.
>
>
> &nbsp;
> #### **Classifier Divergence**
>
> The classifier and the surrogate classifier are not completely disconnected because they are both trained using representations of the (same) feature extractor. The data used to train them is different though, and that is by design, because we want the (main) classifier to have reduced prediction bias. The disagreement between them is of no concern in our view, since their predictions are not used collaboratively for inference; only the (main) classifier is used for inference. Also, note the feature extractor and the surrogate classifier are optimized together, while the classifier is optimized given the representations of the feature extractor. Therefore, the feature extractor does not have to "cope" with both the surrogate classfier and the (main) classifier, but, rather, the classifier has to learn to use the representations of the feature extractor.
>
>
> &nbsp;
> ***
> We want to thank you for you review. Please let us know if any of your points were not addressed properly, or if you have any additional questions.

---

> > ### Comment · Reviewer_yCcF · 2022-11-14
> > **Deeper discussion about SC vs C vs FE during training would help**
> >
> > **Surrogate classifier**: I find the fact that the surrogate classifier is justified only through ablation somewhat unsatisfactory. I would find it useful and certainly of more interest, if there was some more grounded discussion about why this helps (gradient updates?).
> >
> > **Reproducibility**: good
> >
> > **Classifier Divergence**: Right. I guess this relates to my previous concern - more discussion about what happens between the FE and C and SC during the training would help. So that we can understand for why splitting SC and C is actually beneficial and how this plays together with freezing and unfreezing the FE at various stages during the training. So far the paper is somewhat scarce on these.

---

> > > ### Author Response · Authors · 2022-11-15
> > > **Discussion on OBC Components**
> > >
> > > We shortly summarize the motivation behind the design of OBC.
> > >
> > > In order to prevent overfitting the memory data, the feature extractor is trained with both stream and memory data, and is frozen otherwise. The classifier is trained using only memory data, in order to mitigate its prediction bias, and is frozen otherwise.
> > >
> > > Since the feature extractor is trained with the biased data-sampling scheme of experience replay, there needs to be at least one component of the model that "accomodates" this data-sampling bias, and that is why the biased surrogate classifier is necessary. If a surrogate classifier is not used, the feature extractor will be negatively affected by the data-sampling bias (since the classifier is frozen during the first step of OBC - SC; see Section 4.4), and this will decrease learning performance.

---

> ### Author Response · Authors · 2022-11-10
> **Reply to Reviewer yCcF (Part 1/2)**
>
> #### **Definition of Task-Free Continual Learning**
>
> Indeed, as you wrote, task-free continual learning assumes a non-stationary distribution that can change in arbitrary ways. Past work assumes that the model should learn from the data it sees over time, and the evaluation is typically performed after the end of the stream using unseen data that are similar to the ones the model observed.
>
>
> &nbsp;
> #### **Experimental Setup**
>
> Our experimental setup closely follows previous work (e.g., [1, 2]). In particular, we use class-incremental continuums that are presented to the model in an online manner (in small mini-batches of 10 instances). For FashionMNIST and CIFAR-10, we used five binary tasks, for CIFAR-100 ten tasks of ten classes each, and for tinyImageNet, twenty tasks of ten classes each. We will edit our submission to better clarify this point. As you correctly stated, none of the methods we use in our experiments assumes anything, or is informed in any way, about the structure of the stream.
>
> [1] Jin, X., Sadhu, A., Du, J., & Ren, X. (2021). Gradient-based Editing of Memory Examples for Online Task-free Continual Learning. NeurIPS.
>
> [2] Aljundi R., Caccia L., Belilovsky E., Caccia M., Lin M., Charlin L., and Tuytelaars T. (2019). Online continual learning with maximally interfered retrieval. Proceedings of the 33rd International Conference on Neural Information Processing Systems.
>
>
> &nbsp;
> #### **Memory Size and Tractability**
>
> We write that "the larger the stream, the larger the memory will be" as an assumption of what would happen in real-world applications of continual learning. Specifically, we would expect that practitioners would not use memories of a certain size regardless of the size of the stream, but, rather, that larger memories would be used for more lengthy streams, and the opposite.
>
> The computational cost of a method is $O(mn)$ when it needs to process all data in its memory for each incoming stream batch. Such approaches would probably be intractable in real-world applications (with streams that are too massive to store, and memories that would be much larger than the ones used in the research).
>
>
> &nbsp;
> #### **Evaluation**
>
> We purposefully chose datasets that have standard test sets, so that there is no need to generate a test set. The accuracy and the bias are computed with respect to these test sets, after the entire stream has been observed. Hence, these metrics only inform us about the final learning outcome. In contrast, the average information retention is computed by averaging the accuracy on past stream observations at 1000 equally spaced points during continual learning. Therefore, this metric informs us about how well the model is learning throughout the entire stream.
>
>
> &nbsp;
> #### **Novelty and Contributions**
>
> We do not claim that our contribution is breaking down the neural network in a feature extractor and a classifier, nor do we do any explicit past-versus-present sample weighting.
>
> Our paper contrasts what happens when different learning components are trained using both present and memory data, or only memory data. In the former case, there is less feature-extractor overfitting and more prediction bias towards recent observations. In the latter case, there is more feature-extractor overfitting and less prediction bias towards recent observations. We attempt to provide a best-of-both-worlds approach by separately training the feature extractor using both present and replay data (in order to not increase its overfitting) and train the classifier using only replay data (in order to reduce prediction bias). The introduction of the surrogate classifier is done to help facilitate the training of the feature extractor and is motivated by the experimental ablation in Section 4.4.
>
>
> &nbsp;
> #### **Prediction Bias Quantification**
>
> As we state in Section 3.3, the expectations in the computation of prediction bias are with respect to the instances of the unseen test set. There is indeed nothing wrong with using the KL divergence. We chose the JS because we are not strictly trying to approximate a ground truth distribution, but rather try to measure the discrepancy between two vectors of probabilities, so we thought that a symmetric measure makes more sense.
>
>
> &nbsp;
> #### **Comparing Representation Quality**
>
> We argue that, when comparing task-free approaches, representation quality and prediction bias are confounding factors. For instance, approach A learns better representations but is also more biased than approach B, and the evaluation metrics favor approach B. If we correct for the bias in both approaches using our proposed bias-correction approach, we would expect approach A to outperform approach B.

---

> > ### Comment · Reviewer_yCcF · 2022-11-14
> > **Thank you - more questions and comments.**
> >
> > Dear authors, thank you very much for your responses. A couple of follow-up questions a comments:
> >
> > **Definition of Task-Free Continual Learning**: so rather than being agnostic and assuming the distribution can change arbitrarily you actually have a rather strong assumption about the new / unobserved data being generated from some (unknown) mix of all the distributions observed in the past? As you say "somehow similar to the distributions observed in the past" could also mean that the distributions are somehow evolving through the time and that the new / observed data are the most similar to the latest observed distribution. In such a case the bias correction you propose would in my view make little sense, wouldn't it? Is your bias correction therefore closely linked to rather strong assumptions about the importance of past observations in terms of informing the model about the new distributions? And therefore very much not agnostic?
> >
> > **Experimental setup**: Yes, please, make this more clear in the paper. "None of the methods we use assume anything" - I would disagree here. Your bias correction assumes that the past distributions matter are equally (or at least not significantly) less informative for the new data distribution.
> >
> > **Computation complexity**: What would be an example of an O(mn) algorithm in this context?
> >
> > **Evaluation**: What standard test sets are you talking about here? The standard test-sets used for the usual classification purposes (e.g. 60k training, 10k test for FashionMnist https://github.com/zalandoresearch/fashion-mnist)? Or some standard split specifically designed for task-free continual learning? If the first than these sets are inherently constructed with a very strong assumption about the distribution stability across the entire train and test dataset.
> >
> > **Novelty**: As I mentioned in my previous comments, the problem of bias towards recent observation has been noted before. Previous work on this topic should be discussed and it should be made clear, how novel you are in view of these.
> >
> > **Prediction bias quantification**: By my question about expectation I mean *with respect to what probability distribution is the expectation defined*? As in $E_{p(x)} f(x) = \int p(x) f(x) dx$. What is the $p(x)$ (and the variable) in your case?
> >
> > **Representation quality**: Hm ... seems rather vague and certainly not clear from the paper.

---

> > > ### Author Response · Authors · 2022-11-15
> > > **Reply to Follow-Up Questions**
> > >
> > > Dear reviewer,
> > >
> > > thank you for participating in the discussion. Here is our reply:
> > >
> > >
> > > &nbsp;
> > > #### **Definition of Task-Free Continual Learning**
> > >
> > > We repeat the definition from our paper, namely "the online optimization of a model via small minibatches that are sampled from a nonstationary stream." In this setting, the goal is to learn from all observed data equally, despite the nonstationary nature of the stream. As in previous work, the distribution shifts take place only during training, and the model is evaluated after the end of the stream using the standard classification test sets (see for example [1, 2, 3, 4]). (In addition, we also evaluate the model online based on past observations.)
> > >
> > > If there is a distributional mismatch between the training observations and the unseen test data, this becomes a domain adaptation problem, and is outside the scope of the present paper. On the other hand, if, as you wrote, the goal is to only adapt to the "latest observed distribution," this becomes an online learning problem. In continual learning, we want to learn the new and keep remembering the old.
> > >
> > >
> > > &nbsp;
> > > #### **Evaluation**
> > >
> > > Given the answer to your previous question, it hopefully now makes sense why we used the standard classification test sets to compute the accuracy. This is also the case in the relevant literature [1, 2, 3, 4].
> > >
> > > &nbsp;
> > > #### **Distributional Changes in Experimental Setup**
> > >
> > > Distributional changes only take place during training and neither the baselines we use in the paper, nor our proposed approach, assume anything about the way the training distribution changes over time. The only assumption our bias correction approach makes is that all past observations are equally important, an assumption that is central to the evaluation of task-free continual learning [1].
> > >
> > >
> > > &nbsp;
> > > #### **Computational Complexity $O(mn)$**
> > >
> > > An approach that trains a new classifier (output layer) from scratch after every stream minibatch is observed would be of $O(mn)$ computational complexity, and would be intractable in real-world problems. Our method is only $O(n)$.
> > >
> > >
> > > &nbsp;
> > > #### **Novelty and Contributions**
> > >
> > > We do not claim to be the first to observe this bias, and we have cited works that have empirically observed it [5, 6]. For the sake of clarity, we repeat our contributions:
> > > - We are the first to explain why experience replay leads to biased predictions in task-free continual learning;
> > > - We show that the naive approach of only using memory data to train the entire model does reduce prediction bias, but it also leads to a deterioration in learning performance because of overfitting;
> > > - We show that prediction bias can be effectively mitigated after the end of training, by only optimizing the weights of the classifier (final layer of the model) using the data stored in memory;
> > > - We are the first to propose an *online* bias-correction approach that can be applied in a *task-free* scenario. We experimentally show that our proposed approach not only mitigates prediction bias, but also leads to significant increases in final accuracy and average information retention over four different datasets.
> > >
> > >
> > > &nbsp;
> > > #### **Expectation in Prediction Bias**
> > >
> > > The expectation is defined with respect to the data distribution $p(\boldsymbol{x}, y)$ that the test set was sampled from, where $\boldsymbol{x}$ is a data instance and $y$ is its label. This distribution is unknown, therefore, we estimate the expectation by averaging over the instances of the test set (similarly to how the test-set accuracy is computed).
> > >
> > >
> > > &nbsp;
> > > #### **Comparing Representation Quality**
> > >
> > > We will clarify this part of the paper, but we do not think the concept is vague. If two methods have the same memory population algorithm and their classifiers' bias is corrected using the same approach, the only factor that can lead to differences in learning performance is the quality of the representations they learn.
> > >
> > >
> > > &nbsp;
> > > #### **References**
> > > [1] Jin, X., Sadhu, A., Du, J., & Ren, X. (2021). Gradient-based Editing of Memory Examples for Online Task-free Continual Learning. NeurIPS.
> > >
> > > [2] Aljundi R., Caccia L., Belilovsky E., Caccia M., Lin M., Charlin L., and Tuytelaars T. (2019). Online continual learning with maximally interfered retrieval. NeurIPS.
> > >
> > > [3] Wang, Z., Shen, L., Fang, L., Suo, Q., Duan, T., & Gao, M. (2022, June). Improving task-free continual learning by distributionally robust memory evolution. ICML.
> > >
> > > [4] Lee, S., Ha, J., Zhang, D., & Kim, G. (2019, September). A Neural Dirichlet Process Mixture Model for Task-Free Continual Learning. ICLR.
> > >
> > > [5] Pietro Buzzega, Matteo Boschini, Angelo Porrello, and Simone Calderara. Rethinking experience replay: a bag of tricks for continual learning. ICPR.
> > >
> > > [6] Zheda Mai, Ruiwen Li, Hyunwoo Kim, and Scott Sanner. Supervised contrastive replay: Revisiting the nearest class mean classifier in online class-incremental continual learning.

---

> > > > ### Comment · Reviewer_yCcF · 2022-11-20
> > > > **Valuable explanations, paper much clearer now**
> > > >
> > > > Dear authors,
> > > > Thanks a lot for your responses and for all the clarifications. These are really good and I would strongly encourage you to include these into the paper. They would very much improve the clarity and therefore quality of your paper, preventing confusion and misunderstandings. When you do, I would be happy to increase my rating of your paper.
> > > > While the setup may be clear to you,  the current wording (e.g. "setting is completely agnostic to the way the distribution changes over time") raises doubts. In contrast, what you say in **Definition of Task-Free Continual Learning**, **Evaluation** and **Distributional Changes in Experimental Setup** is completely clear and I fully agree with the distinction and delimination between domain adaptation and online learning you do there, it would help the readers a lot to see these in the paper directly.
> > > > The same is true **Prediction Bias** and **Representation Quality** - please make the respective sections in your paper less opaque by using your own explanations here. They are very helpful.

---

> > > > > ### Author Response · Authors · 2022-11-23
> > > > > **Thank You**
> > > > >
> > > > > Dear reviewer,
> > > > >
> > > > > thank you for your reply. Unfortunately, it is not currently possible to update the paper because the deadline for that (November 18) had already passed before your latest reply. We have already included some of these additional explanations from our rebuttal in our revised submission (please refer to the Revision Summary at the top of the comments). If you feel like your concerns have been addressed by our discussion, we kindly ask if you could reconsider your score, and we assure you that we will integrate all the remaining additional explanations from our discussion in the final version of our paper. We want to thank you again for your review and your valuable suggestions.

---

### Official Review · Reviewer_S1Ps · 2022-10-25

**Confidence:** 4
**Correctness:** 4
**Technical Novelty And Significance:** 4
**Empirical Novelty And Significance:** 4
**Recommendation:** 8

**Clarity, Quality, Novelty And Reproducibility:**

The paper is easy to read and follow. The ideas in the paper are clear and presented in a coherent manner. The contribution seems to be novel. The authors have pledged to make the source code public upon acceptance for reproducibility.

**Strength And Weaknesses:**

Strengths:
1. Online task-free continual learning has recently drawn more attention because it focuses on solving practical challenges. The direction of this paper is promising as it focuses on correcting the bias incurred during training, addressing a major problem in this area.
2. The claim of correcting the prediction bias by changing the way the classifier is trained is well supported by extensive experimentation of various methods on a good amount of datasets.
3. The proposed method, Online Bias Correction, is simple but very effective. Also, it acts as a wrapper around other task-free continual learning methods. The main advantage of this is that it can be applied to existing methods without any major change in existing architecture and training procedure.
4. The paper also includes the ablation study on surrogate classifier which was introduced by the proposed method. Its effectiveness is well supported by extensive experimentation.

Weaknesses:
1. Even though the paper has used a variety of datasets, many recent papers in this area also use CORe50 [1]. I believe that it would strengthen the experiments and would make it easier to compare against some other work.
2. In section 3.2, the explanation on data sampling bias is unclear. First, they state that past and future observations have contributed equally. Then it is changed to not contributing equally, making it confusing. I believe that a clear mathematical definition would likely remove the confusion.
3. Main concern of the method is that it is a wrapper over the existing methods which makes it dependent on them and also might not guarantee effectiveness on the methods outside the ones used in the experiments.
4. Table 1 includes a column called “Bias”. Is this the result of the Jensen-Shannon divergence?

[1] Lomonaco, Vincenzo, and Davide Maltoni. "Core50: a new dataset and benchmark for continuous object recognition." Conference on Robot Learning. PMLR, 2017.


**Summary Of The Paper:**

The paper focuses on online bias correction during task-free continual learning. The paper first shows, both theoretically and empirically, why simple experience replay biases on the results of the recent stream observation. Second, the paper introduces the metric to quantify prediction biases. Using the observation from this, the paper proposes a simple approach called Online Bias Correction which appropriately modifies the final layer of the network to correct for the biases online. Finally, the paper concludes with extensive experimentation of the OBC showing significant improvement on a number of task-free continual learning methods on multiple datasets.

**Summary Of The Review:**

The paper focuses on a problem that is underexplored. The ideas presented are novel. The contribution is significant in the area of Bias correction, especially in online settings of task-free continuous learning. Even though the proposed method is a  wrapper around the existing methods, I think it is a significant and valuable contribution in this area as it is also evident from the rich set of results included.

---

> ### Author Response · Authors · 2022-11-10
> **Reply to Reviewer S1Ps**
>
> #### **Comparison on CORe50**
>
> Thank you for the suggestion. Since CORe50 is a temporally coherent dataset, we believe that it will be indeed valuable to include a comparison on it. We will update our submission within the next few days to include the results.
>
>
> &nbsp;
> #### **Data Sampling Bias**
>
> We will explain how this bias arises using a numerical example. Consider a learner that has previously observed $990$ data instances, and these instances are stored in the learner's memory (let's assume it is infinite for simplicity's sake). A new minibatch of $10$ instances is now observed by the learner. If the learner wants to update the model in an unbiased way, all $990 + 10$ data instances should be equally likely to participate in this update. Assuming that $20$ instances will be used in the update, the probability that any instance participates is therefore $20 / (990 + 10) = 0.02$. In experience replay, however, the $10$ instances of the new minibatch would participate in the update with a probability of $1$, while $10$ instances would be randomly sampled from memory, hence each memory instance is sampled with a probability of $10 / 990 \simeq 0.01$. Therefore, in experience replay, the probability that a new instance will participate in the update is $50$ times larger than in the unbiased case, while the probability that any memory instance will participate in the update is approximately half of what it would be in the unbiased case. The confusion was probably caused because we discuss both about minibatches contributing equally (in experience replay) and also about samples contributing equally (in the unbiased case). We will edit our submission to clarify it further.
>
>
> &nbsp;
> #### **The Proposed Approach is a Wrapper over Other Methods**
>
> This is indeed something we acknowledge within the paper, but we consider it only a minor limitation given how widespread the experience-replay paradigm is in task-free continual learning.
>
>
> &nbsp;
> #### **Bias Column in Tables**
>
> Indeed, the numbers in the columns titled "Bias" are computed using the JS divergence in Eq. (2).
>
>
> &nbsp;
> ***
> We want to thank you for you review. Please let us know if any of your points were not addressed properly, or if you have any additional questions.

---

> > ### Comment · Reviewer_S1Ps · 2022-11-29
> > **Acknowledgement**
> >
> > Thank you for the clarifications. I am confirming the initial rating since this paper is focussing on a relevant and underexplored problem.

---

### Official Review · Reviewer_zZ8e · 2022-10-25

**Confidence:** 4
**Clarity, Quality, Novelty And Reproducibility:** Clear and should be easily easy to re…
**Correctness:** 3
**Technical Novelty And Significance:** 3
**Empirical Novelty And Significance:** 2
**Recommendation:** 6

**Strength And Weaknesses:**

Strengths
- A very simple but highly effective method for bias correction in task-free continual learning
- Ablates the effect of various components (e.g. utility of surrogate classifier).
- Well written paper with decent set of evaluations in the vision space.
- Good comparison to baselines and shows particularly good performance with respect to the bias metric proposed, and another metric proposed by Cai et al (2021).

Weaknesses
- Lack of understanding into why various components work despite ablations. This makes it difficult to find as much value in the paper, given the focus on toy datasets. Concretely, in section 4.5 it details why a surrogate classifier is necessary. However, there is no exposition on why surrogate classifier is necessary. There is a reference to gradient flow and figure 1 but nothing beyond that.

Minor Comments & Questions
- Provide a name for the bias metric, and then use that name in the tables
- Highlight performance of OBC in tables.

**Summary Of The Paper:**

Paper proposes a method to correct a recency bias in replay-based task-free continual learning, but separately optimising the final connected layer  of network from the rest of the network. Focuses on continual learning in vision with evaluations in the area.

**Summary Of The Review:**

A simple method to mitigate the effect of the recency bias in task-free continual learning. Well written paper. Accept.

---

> ### Author Response · Authors · 2022-11-10
> **Reply to Reviewer zZ8e**
>
> #### **OBC Components and Motivation**
>
> In Section 3.2, we explained why the traditional experience-replay approach of combining current observations and an equal number of replay instances biases the predictions of the model towards the current observations. Next, in Section 3.4, we show that after the stream has ended, we can mitigate this prediction bias by updating only the weights of the classifier (final layer) of the model using only the data in memory. In Section 3.4, we also showed that if the entire model (both the feature extractor and the classifier) is trained exclusively using replay data throughout the continuum (this is the MRO approach), then the prediction bias is mitigated but the model has inferior performance due to overfitting. Our goal was to take advantage of all this information, in order to propose an approach that does online bias correction without increased overfitting.
>
> The main components of our OBC approach are two: i) training the feature extractor using both current observations and replay data and the classifier using only replay data; ii) introducing a surrogate classifier to facilitate the training of the feature extractor. The first component is motivated by the information we summarized in the previous paragraph. The use of the second component is motivated by the experimental results in Table 5. Specifically, we saw that using a biased surrogate classifier when training the feature extractor leads to significantly increased accuracy and average information retention, and to significantly decreased prediction bias. (We write that the surrogate classifier is biased because it is trained with both current observations and replay data.)
>
>
> &nbsp;
> #### **Toy Datasets**
>
> The datasets in our experiments have been used extensively in the continual learning literature. Please keep in mind that in a task-free continual setting, there is only one pass performed over the stream, and only a small fraction (1-5% in our experiments) of the data is available for training the model at each moment in time. As can be seen from our results, CIFAR-100 and tinyImageNet are extremely challenging in this setting.
>
>
> &nbsp;
> ***
> We want to thank you for you review. Please let us know if any of your points were not addressed properly, or if you have any additional questions.

---

### Author Response · Authors · 2022-11-19
**To All Reviewers: Contributions and Revision Summary**

Dear reviewers,

we would like to once again thank you for your reviews and your suggestions. Here, we repeat the contributions of our work, and we list the changes that we have made in the revised version of the paper.

Our contributions are the following:
1. We are the first to provide a concrete explanation of why experience replay leads to predictions that our biased with respect to recent observations. (Previous works had observed the existence of the bias, but did not specify the mechanism.)
2. We propose a metric to quantify how biased the predictions of a model are, with respect to an unseen test set.
3. We propose a simple Online Bias Correction (OBC) approach that can be applied online in task-free settings (that is, without any assumptions on how the data distribution changes during training). We experimentally show that OBC, not only mitigates the prediction bias of approaches that follow the experience-replay paradigm, but also that it leads to significant increases in accuracy and information retention. The results are consistent among all five datasets used in the paper (including CORe50).

Changes in the revised version of the paper:
- As suggested by Reviewer S1Ps, we present results on the CORe50 dataset in the appendix. These results show that OBC consistently reduces prediction bias, and also increases accuracy and average information retention. The hyperparameters of the experiment are given in the appendix.
- We added the Prediction Bias results from the experiments of Section 4.2 in the appendix. We did not include them in the main paper because the resulting table has 13 columns and would not be readable in print. For all method-dataset combinations, using OBC leads to significant reductions in prediction bias.
- We rewrote the description of data-sampling bias (Section 3.2), and provided a numerical example in the appendix
- We described the motivation behind the design and components of OBC in the last papragraph of Section 3.5.
- We added a segment in Section 4.1 stressing that our experimental setup follows previous work and included references.
- We extended the discussion in Section 4.5 for the sake of clarity.
- Due to lack of space, we moved the hyperparameter sensitivity section to the appendix.

---

### Decision · Program_Chairs · 2023-01-20

**Decision:**

Accept: poster

**Justification For Why Not Higher Score:**

Lack of conceptual justification or intuitive motivation for why this method should work.

**Justification For Why Not Lower Score:**

Paper identifies and tackles an important problem in continual learning w/ experience replay and proposes an effective method.

**Metareview: Summary, Strengths And Weaknesses:**

The paper proposes a method to correct recency bias in replay-based task-free continual learning by optimizing the final connected layer of network from the rest of the network and demonstrates its effectiveness on a number of vision benchmark tasks.

The reviewers agreed that the paper points out an important problem in experience replay relevant to continual learning. The reviewers appreciated the simplicity of the method and agreed that the empirical results are strong.

The reviewers thought the paper could benefit greatly from more motivation, intuition, and conceptual justification for the design choices of the proposed method, and further clarification of various definitions such as what "bias" means in this context.

**Note From Pc:**

if the above contains the word "oral" or "spotlight" please see: "oral" presentation means -> notable-top-5% and "spotlight" means -> notable-top-25%. As stated in our emails, we are disassociating presentation type from AC recommendations

**Summary Of Ac-Reviewer Meeting:**

zZ8e and cbbB attended. The rest of the reviewers were unresponsive.

Notes:

zZ8e: The paper was clarified during the author response. However, didn’t think the contribution was quite significant.

cbbB: Not against accepting, but contributions are not that strong. Concern was about the clarity of the setup. They work on very general setup in which they don’t assume any distribution. They don’t really defined no-stationarity. Raised concern about some learnability issue, and there were some discussion. At the end was convinced even in the literature there’s no constant setup. They just follow the standard setup. I feel I shouldn’t blame too much. Another concern was definition of bias. They just compare the difference between the two in the class proportion, it’s not consistent with the standard definition of bias. Not about the average over draw of training samples, but they just take average over the given training samples.

zZ8e: I agree the term bias is a bit loaded.

cbbB: They only compare P(y) instead of P(y|x) being unbiased. Some concerns about the model of bias, not comprehensive enough but the experiments seem OK. Conceptually is not satisfying.
Last concern is how they correct bias. It’s not very convincing. They just freeze the feature extractor and retrain the last layer using memory mini batch. I don’t know if there are any guarantees this correctly cancels the bias that’s introduced by the experience replay.

zZ8e: No theory but experimental performance holds up. Experience replay relatively studied… in terms of this method they just used existing experience replay and drop their method on top of that. This simplicity is a good thing.

My concern is that they could have clarified further certain parts of the paper or intuition is left to the imagination. However this may not be enough reason to reject.